# IRAK2, an Immune and Radiation-Response Gene, Correlates with Advanced Disease Features but Predicts Higher Post-Irradiation Local Control in Non-Metastatic and Resected Oral Cancer Patients

**DOI:** 10.3390/ijms24086903

**Published:** 2023-04-07

**Authors:** Chih-Chia Yu, Hon-Yi Lin, Chen-Hsi Hsieh, Michael W. Y. Chan, Wen-Yen Chiou, Moon-Sing Lee, Chen-Lin Chi, Ru-Inn Lin, Feng-Chun Hsu, Liang-Cheng Chen, Chia-Hui Chew, Hsuan-Ju Yang, Shih-Kai Hung

**Affiliations:** 1Department of Medical Research, Dalin Tzu Chi Hospital, Buddhist Tzu Chi Medical Foundation, Chia-Yi 62247, Taiwan; hippo67117@yahoo.com.tw; 2Department of Radiation Oncology, Dalin Tzu Chi Hospital, Buddhist Tzu Chi Medical Foundation, Chia-Yi 62247, Taiwan; doc31221@gmail.com (H.-Y.L.); cwyncku@gmail.com (W.-Y.C.); rtjo5566@gmail.com (M.-S.L.); whitefly@gmail.com (R.-I.L.); jaine0318@yahoo.com.tw (F.-C.H.); atonsobek@yahoo.com.tw (L.-C.C.); chiahui_c@yahoo.com (C.-H.C.); dl60885@tzuchi.com.tw (H.-J.Y.); 3School of Medicine, Tzu Chi University, Hualien 97004, Taiwan; 4Division of Radiation Oncology, Far Eastern Memorial Hospital, New Taipei City 220216, Taiwan; chenciab@gmail.com; 5Institute of Traditional Medicine, School of Medicine, National Yang Ming Chiao Tung University, Taipei 11221, Taiwan; 6School of Medicine, National Yang Ming Chiao Tung University, Taipei 11221, Taiwan; 7Department of Biomedical Sciences, National Chung Cheng University, Min-Hsiung, Chia-Yi 62102, Taiwan; biowyc@ccu.edu.tw; 8Epigenomics and Human Disease Research Center, National Chung Cheng University, Min-Hsiung, Chia-Yi 62102, Taiwan; 9Center for Innovative Research on Aging Society (CIRAS), National Chung Cheng University, Min-Hsiung, Chia-Yi 62102, Taiwan; 10Research Center for Environmental Medicine, Kaohsiung Medical University, Kaohsiung 80708, Taiwan; 11Department of Pathology, Chiayi Chang Gung Memorial Hospital, Chia-Yi 61363, Taiwan; skypoem0038@gmail.com

**Keywords:** biomarker, IRAK2, oral squamous cell carcinoma, radiotherapy, local control

## Abstract

Gene Ontology (GO) analysis can provide a comprehensive function analysis for investigating genes, allowing us to identify the potential biological roles of genes. The present study conducted GO analysis to explore the biological function of IRAK2 and performed a case analysis to define its clinical role in disease progression and mediating tumor response to RT. Methods: We performed a GO enrichment analysis on the RNA-seq data to validate radiation-induced gene expression. A total of 172 I-IVB specimens from oral squamous cell carcinoma patients were collected for clinical analysis, from which IRAK2 expression was analyzed by immunohistochemistry. This was a retrospective study conducted between IRAK2 expression and the outcomes of oral squamous cell carcinoma patients after radiotherapy treatment. We conducted Gene Ontology (GO) analysis to explore the biological function of IRAK2 and performed a case analysis to define its clinical role in mediating tumor response to radiotherapy. GO enrichment analysis to validate radiation-induced gene expression was performed. Clinically, 172 stage I-IVB resected oral cancer patients were used to validate IRAK2 expression in predicting clinical outcomes. GO enrichment analysis showed that IRAK2 is involved in 10 of the 14 most enriched GO categories for post-irradiation biological processes, focusing on stress response and immune modulation. Clinically, high IRAK2 expression was correlated with adverse disease features, including pT3-4 status (*p* = 0.01), advanced overall stage (*p* = 0.02), and positive bone invasion (*p* = 0.01). In patients who underwent radiotherapy, the IRAK2-high group was associated with reduced post-irradiation local recurrence (*p* = 0.025) compared to the IRAK2-low group. IRAK2 plays a crucial role in the radiation-induced response. Patients with high IRAK2 expression demonstrated more advanced disease features but predicted higher post-irradiation local control in a clinical setting. These findings support IRAK2 as a potential predictive biomarker for radiotherapy response in non-metastatic and resected oral cancer patients.

## 1. Introduction

Oral squamous cell carcinoma (OSCC) ranks sixth in cancer incidence in Taiwan, and radiotherapy (RT) is one of the essential treatment modalities for managing OSCC patients [1]. RT is mainly applied with surgery in advanced-stage patients [1,2]. However, local recurrence and distant metastasis are still major clinical problems that lead to poor overall survival in oral cancer patients [2,3]. Although some clinical and pathological factors help to identify patients at high risk for cancer recurrence, there is a need to identify reliable molecular biomarkers to predict clinical outcomes in OSCC patients.

RNA sequencing (RNA-seq) has been of great significance for finding novel biomarkers in radiotherapy. It is a powerful way to obtain transcriptomic information from organisms within clinical phenotypes and understand molecular feature alterations in clinical prognosis and therapy [4]. Bioinformatics research depends on high-quality databases which have been widely used in studying cancer-related genes and are considered promising tools for molecular prediction [5]. Gene Ontology (GO) functional analysis, as the vital application of RNA-Seq data analysis, provides insights into the differentially expressed genes’ biological processes and molecular functions [4]. Thus, the GO functional analysis of transcriptomic data can offer an objective screening approach to identify biomarkers for predicting and monitoring radiation efficacy [4,6]. However, reliable studies that focus on changes in gene expression during radiotherapy treatment based on bioinformatics are scarce.

Interleukin-1 receptor-associated kinase 2 (IRAK2) is a critical regulator of interleukin-1 receptor (IL-1R)/toll-like receptor (TLR)-mediated inflammation and is involved in the nuclear factor-κB (NF-κB) and mitogen-activated protein kinase (MAPK) signaling pathways [7]. It is also recognized to participate in the apoptotic response through multiple mechanisms [8,9,10]. Recently, the role of IRAK2 has been investigated in various cancers. IRAK2 is highly expressed in pancreatic cancer patients and associated with poor prognosis. IRAK2 knockdown has led to a significant impairment of pancreatic cancer cell proliferation [11]. IRAK2 has also been found to be expressed in triple-negative breast cancer cells, and its downregulation decreases tumor growth [12]. A previous study has demonstrated an association between IRAK2 expression and post-radiotherapy (post-RT) outcomes [10]; however, its clinical application is limited because only a specific population (i.e., only early-stage I–II patients) was enrolled. Hence, it is crucial to conduct a further study which includes all clinical populations. 

To elucidate if IRAK2 plays a crucial functional role in mediating the radiation-induced responses of OSCC, in the present study, we used GO classification to validate the IRAK2-associated molecular network and biological processes to ensure it is an effective prediction marker. Moreover, the present study expanded the patient population to all non-metastatic and resected patients (including stage I–IVB) to offer a potential clinical application. 

## 2. Results

### 2.1. Investigating Irradiation Response Genes in Irradiated OML1 Oral Cancer Cells through Transcriptome Analysis

In the present study, we utilized RNA-seq data and focused on the post-irradiation upregulated gene profiles in the oral cancer cell line OML1. After irradiation, 25 genes were significantly upregulated based on the selection criterion of a log2 fold change (FC) of >1.4 with a *p*-value of <0.01 (Figure 1). Figure 1 shows the distribution of the gene expression between untreated and ionizing radiation (IR)-treated OML1 cells. 

### 2.2. Gene Ontology Analysis Shows Enriched Biological Processes of Post-Irradiation (Post-IR) Upregulated Genes

As derived from the volcano plot in Figure 1, Figure 2 depicts the expression levels of the 25 post-IR upregulated genes, including *PHLDA2*, *HBEGF*, *IRAK2*, *MCAM*, *CSF2*, *CA2*, *INHBA*, *LY6G5B*, *INHBB*, *HOXC13*, *AC007405.6*, *MEF2BNB-MEF2B*, *ACTR3C*, *HSD3B7*, *ECM1*, *RP11-81N13.1*, *AC074091.13*, *LYSMD1*, *FABP3*, *C15orf59*, *AC099342.1*, *CD74*, *C4B*, *FRMD3*, and *KIAA0408*. Remarkably, we found that IRAK2 was one of the top three significantly upregulated genes after irradiation. Subsequently, we used GO enrichment analysis to classify the selected genes, focusing on their biological functions and processes. We observed that the 25 genes were involved in signaling various cell regulatory processes, such as cell communication, migration, apoptosis, proliferation, and stress responses (e.g., to chemicals, lipids, and oxygen). In addition, several genes were found to participate in the immune and inflammatory response, including cytokine-mediated signaling pathways and the regulation of immune system processing (Figure 3). Significant GO enrichment profiles for the biological processes are summarized in Table 1. IRAK2 plays multiple roles in 10 of the top 14 GO-enriched biological processes, including immune regulation and stress response.

### 2.3. IRAK2 Expression Predicts Local Control for Patients Who Received Radiotherapy 

Correlations between IRAK2 expression and the clinical features of OSCC patient samples are summarized in Table 2. The results indicated that the high expression of IRAK2 correlated statistically significantly with pT3-4 (*p* = 0.01), advanced pathology stage (*p* = 0.02), and positive bone invasion (*p* = 0.01). Among irradiated patients, the high IRAK2-expressed group showed better local control than those with low IRAK2 expression (Figure 4A, *p* = 0.025). In contrast, no local control difference according to IRAK2 expression status was observed in non-irradiated patients (Figure 4B, *p* = 0.54). For patients treated with RT, Cox proportional hazard regression confirmed this observation in univariate (HR, 0.301; 95% CI, 0.098–0.923; *p* = 0.036; Figure 4C) and multivariate analyses (HR, 0.243; 95% CI, 0.071–0.838; *p* = 0.025; Figure 4D). In multivariable analysis, we adjusted age, gender, pathological stage, radiotherapy dose, chemotherapy, the status of surgical margin, and the expression level of IRAK2 (Table 3). We found that IRAK2 expression (HR = 0.26, 95% CI = 0.10–0.71, *p* = 0.008) was statistically significantly associated with local recurrence (*p* = 0.008), while chemotherapy status was not (*p* = 0.09). However, the outcome benefit of local control did not translate into cancer-cause-specific and overall survival (Figure 4E,F and Table 4) in both the RT and no-RT groups. These results suggest that high IRAK2 expression correlates with advanced disease features but has higher local control in OSCC patients treated with RT. 

## 3. Discussion

Radiotherapy is an important treatment modality for OSCC patients. However, unfortunately, the therapeutic outcomes are not fully satisfactory. Regional recurrence and/or distant metastasis remain a significant hurdle for radiotherapy. The radiation-induced response is a complicated pathophysiological phenomenon involving many biological and genetic alterations that are responsible for the recurrence of cancer and metastasis following radiotherapy [1]. Hence, finding useful predictive biomarkers to estimate OSCC response to radiotherapy is crucial. The use of bioinformatics technology to mine RNA-seq data has been widely used to analyze disease-related differentially expressed genes to compare the significance between different gene signatures and explore effective biomarkers, and also to understand the biology underlying the association by performing functional studies of the candidates that we identified [13]. GO functional analysis is quite a powerful bioinformatics initiative that can help gain a comprehensive and deeper understanding of the biological functions and interactions of the investigated genes [4]. GO functional analysis helps identify an accurate biomarker for the radiation response to assess clinical appropriateness. It is well known that RT modulates the immunomodulatory response [14,15,16], and the synergy between RT and the immune system is currently receiving significant attention [17,18,19]. The data highlight the utility of IRAK2, which effectively predicts the radiation response. In this study, we utilized GO analysis to validate radiation-induced gene expression and provide complete information about IRAK2. The top 25 upregulated genes were identified in IR-treated cells, indicating that the essential enriched pathways were mainly involved in regulating stress-related and immune system functions. Importantly, we verified that IRAK2 displayed significant upregulation after irradiation. GO analysis identified that IRAK2 is tightly correlated with 10 of the top 14 enriched biological processes; this finding suggests it is closely related to cancer immunity and demonstrates the significant role of IRAK2 in the cellular response to irradiation.

The clinical value of targeting IRAK gene family members, specifically IRAK1 and IRAK4, has been elucidated in several cancer types [20,21,22]. For example, IRAK1 was upregulated in hepatocellular carcinoma (HCC) tissue. High IRAK1 expression was associated with large tumor size, metastasis, advanced T status, and poorer overall survival (OS) in HCC patients [20]. Similarly, activated IRAK4 (p-IRAK4) was associated with poor prognosis in colorectal cancer (CRC) patients at stage IIb–IV more notably than in patients at stage I-II [22]. In addition, IRAK2, an essential IL-1R/TLR signaling mediator, has been recognized to regulate immune response [7]. IRAK2 has been implicated in playing several roles in human cancers. For example, IRAK2 mediates the phosphorylation of Smurf1, which triggers ER stress-mediated apoptosis in colorectal cancer cells [23]. Another previous study showed that the SNP rs779901 T allele genotype in IRAK2 was associated with the increased expression of IRAK2 mRNA in non-small cell lung cancer (NSCLC) patients [24]. However, few studies have investigated the clinical significance of IRAK2 in OSCC patients. In the present study, we included all non-metastatic and resected oral OSCC patients in our analysis. We found that a high IRAK2 expression correlated with advanced clinical features, including pT3-4 status, overall pathological III-IVB stage, and positive bone invasion.

In terms of treatment response, a high expression of p-IRAK4 has been reported to correlate with a poor chemotherapy response [22]. A poor response was also found in other different solid tumors [25,26,27]. Recently, a member of the IRAK family, IRAK1, was also implicated in radiation response [28,29]. It was shown to promote radioresistance in zebrafish models [28]. Interestingly, we also found that high IRAK2 expression (HR = 0.26, 95% CI = 0.10–0.71, *p* = 0.008) was statistically significantly associated with local recurrence. However, the high-IRAK2-expression group showed better local control than the group with low IRAK2 expression among irradiated patients. These findings imply that high-IRAK2-expression patients should be considered for radiotherapy as an adjuvant treatment. Clinically, IRAK2 expression can help to clarify which patients need to receive radiotherapy. Taken together, IRAK2 correlates with advanced disease features but has higher post-irradiation local control in clinical settings.

These findings support the possible outcome-predicting value of IRAK2. However, we found that the observed local control benefit did not translate into overall survival. There may be several reasons for this observation. First, the patient number is limited. This factor may mask the prognostic values of IRAK2. Second, this was a retrospective study, and several patient and tumor characteristics could not be controlled effectively. Third, we included OSCC patients ranging from stage I to IVB. The complicated treatment modalities might influence the analysis. For example, most treatment failure of oral cancer is due to local recurrence. Surgery is still effective as salvage therapy, especially in early oral cancer. These kinds of groups would decrease the prognostic value of survival by IRAK2. These weak points need further investigation. Another concern is how to find an optimal threshold value of IRAK2. Initially, we tried to use the receiver operating characteristic (ROC) curve with the Youden index to find an optimal threshold value of IRAK2. However, we did not find effective differences by using the ROC-estimated optimal cutoff point. The ROC curve assumes that the classes are balanced. Some level of imbalanced classes, such as age, gender, or cancer stage, may impair the ROC’s performance. We used the median IRAK2 value as a cutoff value and the Kaplan–Meier curve to test survival. The Cox proportional hazards regression model was then used to adjust multiple factors. We obtained statistically significant results. These methods seemed more appropriate to the ROC curve in this study. For effective clinical applications, further prospective studies are warranted.

In summary, the present study applied GO analysis to expand upon RNA-sequencing-based data to conduct functional biological analysis to further verify the molecular functions of IRAK2, which may exhibit changes correlating with the radiation response in stage I-IVB OSCCs. Our study demonstrated that high IRAK2 expression predicted higher post-irradiation local control. These findings support IRAK2 as a potential radiation biomarker. However, the detailed clinical significance and mechanisms by which IRAK2 triggers post-IR cellular responses still need to be fully resolved. Thus, additional studies are required to evaluate further the mechanisms involved in the relationship between IRAK2 and the irradiation response, and prospective studies are warranted to validate the clinical applications.

## 4. Materials and Methods

### 4.1. Clinical Specimens

As mentioned previously [30], post-operative RT with or without concurrent Cisplatin-based chemotherapy was prescribed for resected oral cancer patients with the indicated adverse features. RT indications were pT3-4, pN+, and positive or close surgical margins (i.e., ≤1 mm). We used volumetric-modulated arc therapy (VMAT) to prescribe irradiation with the following doses: 60–72 Gy to the primary surgical bed, 60–66 Gy to the high-risk nodal basins, and 50–60 Gy to the low-risk nodal basins. Electronic portal imaging and cone-beam tomography were conducted weekly during the irradiation course. The conventionally fractionated dose was delivered, i.e., 1.8–2 Gy per day, five days per week, with 6 MV photons. 

Cisplatin-based chemotherapy was concurrently given as a radiosensitizer in patients with additional high-risk adverse features, such as positive/close surgical margins and extracapsular nodal extension. The prescribed dose was 80–100 mg/m^2^ administered every three weeks, in two to three cycles during RT, depending on the patient’s condition. All OSCC formalin-fixed paraffin-embedded histological samples were examined retrospectively in an anonymous de-identified manner. 

### 4.2. Functional Analysis of Radiation-Responsive Genes

We had previously conducted RNA-seq in the OML1 human oral carcinoma cell line. These data were deposited in the Gene Expression Omnibus (GEO) database (GSE165585). Gene Ontology (GO) enrichment analysis was performed to classify the functions of significantly expressed genes using the Database for Annotation, Visualization, and Integrated Discovery (DAVID) (https://david.ncifcrf.gov/, accessed on 11 August 2021). Genes of enriched biological processes and GO terms using the functional annotation tool were identified according to the instructions of the DAVID manual.

### 4.3. Definition the High or Low Scores of IRAK2

We used immunohistochemistry to stain the formalin-fixed paraffin-embedded patient samples for IRAK2 analysis. Oncological pathologists defined protein expression scores using the multiplied values of stained intensity (0–3) and percentages (0–100%), rating 0–300. The median IRAK2 value was applied as a cutoff value to differentiate high or low expression. 

### 4.4. Research Database of Clinical Outcomes

Data were collected through the Prospective-Coding Cancer Registry Database in Dalin, Taiwan, a regular national audit cancer database for oncological statistics and research. Patient features were analyzed according to relevant demographic and clinical factors, such as age, gender, pathologic grade, pathologic stage, pathological tumor (pT), pathologic lymph Node (pN), surgical margin, RT, CT, CCRT, treatment, bone invasion, extracapsular spread of lymph node, lymphatic permeation, vascular permeation, perineural invasion, submandibular gland invasion, and skin invasion. The tumors were staged according to the 7th American Joint Committee on Cancer (AJCC) TNM classification. Pearson correlation was applied to evaluate correlations between clinical–pathological parameters and IRAK2 expression. 

### 4.5. Statistical Analysis

All statistical analyses were performed using SigmaPlot software, version 10.0 (Systat Software Inc., San Jose, CA, USA) and SPSS (version 12.0; SPSS Inc., Chicago, IL, USA). Continuous data were presented as mean ± standard deviation, and their statistically significant levels were calculated using Student’s *t*-test. Categorical data were analyzed using the Chi-square test. Time-to-event endpoints were estimated using the Kaplan–Meier plot, and the log-rank test was applied to assess curve differences between groups. The Cox proportional hazards regression model was applied for univariate and multivariate analysis. All hazard ratios were provided with 95% confidence intervals to delineate adequate size. *p*-values of less than 0.05 were defined as statistically significant.

## 5. Conclusions

The present study conducted GO analysis to expand RNA-sequencing-based data into a functional biological analysis in all non-metastatic and resected OSCCs. Biologically, we found that IRAK2 plays a crucial role in the radiation-induced response, including immune regulation and stress response. Clinically, patients with a high expression level of IRAK2 demonstrated more advanced disease features but predicted higher post-irradiation local control. These findings support IRAK2 as a potential predictive biomarker for RT response in non-metastatic and resected OSCC patients.

## Figures and Tables

**Figure 1 ijms-24-06903-f001:**
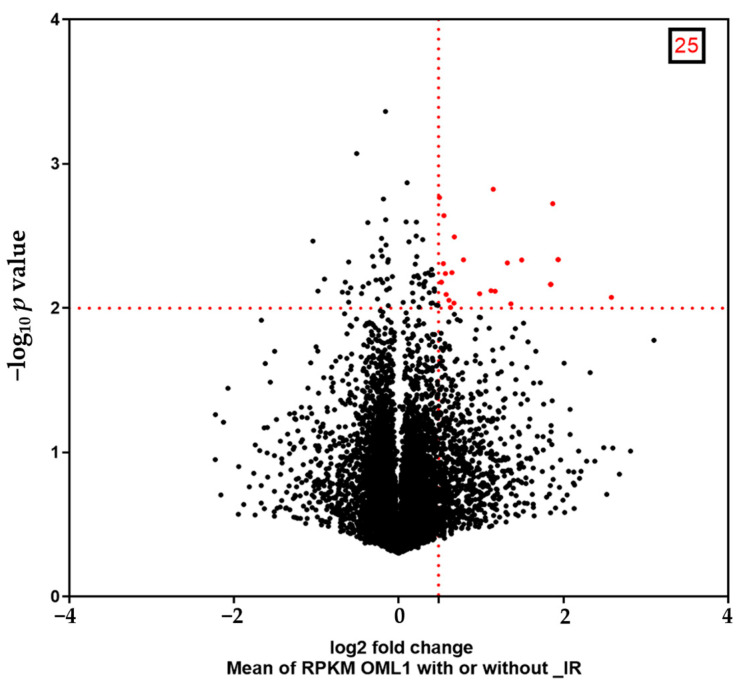
Volcano plot illustrates genes differentially expressed in untreated and ionizing radiation (IR)-treated OML1 cell sub-lines. Twenty-five genes are statistically significantly upregulated (red). The *x*-axis is the log2 fold change, and the *y*-axis is the log10 *p*-value. Abbreviation: RPKM, reads per kilobase per million mapped reads.

**Figure 2 ijms-24-06903-f002:**
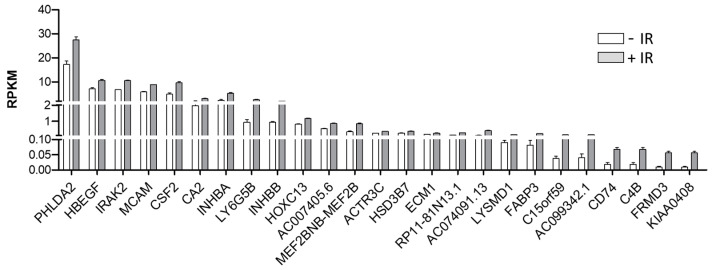
IRAK2 is induced by ionizing radiation in OML1 cells. Expression of post-irradiation upregulated genes. The mean expression of the 25 most significantly upregulated genes in IR-treated OML1 cells compared to non-treated cells. The plot shows the RPKM values for each transcript in two independent replicated experiments.

**Figure 3 ijms-24-06903-f003:**
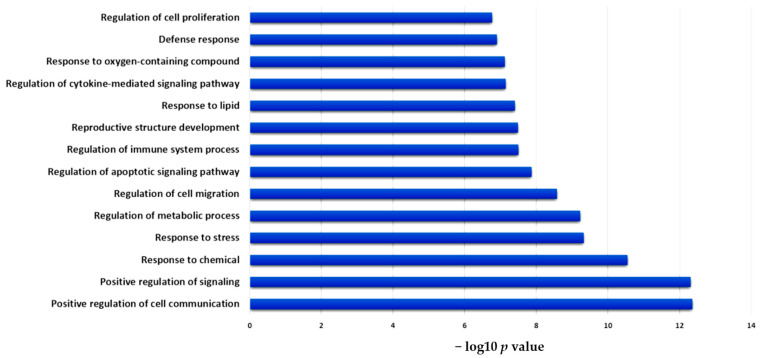
Functional annotation of significantly enriched GO categories of upregulated genes. Bar charts show the top 14 enriched Gene Ontology (GO) categories for post-irradiation biological processes.

**Figure 4 ijms-24-06903-f004:**
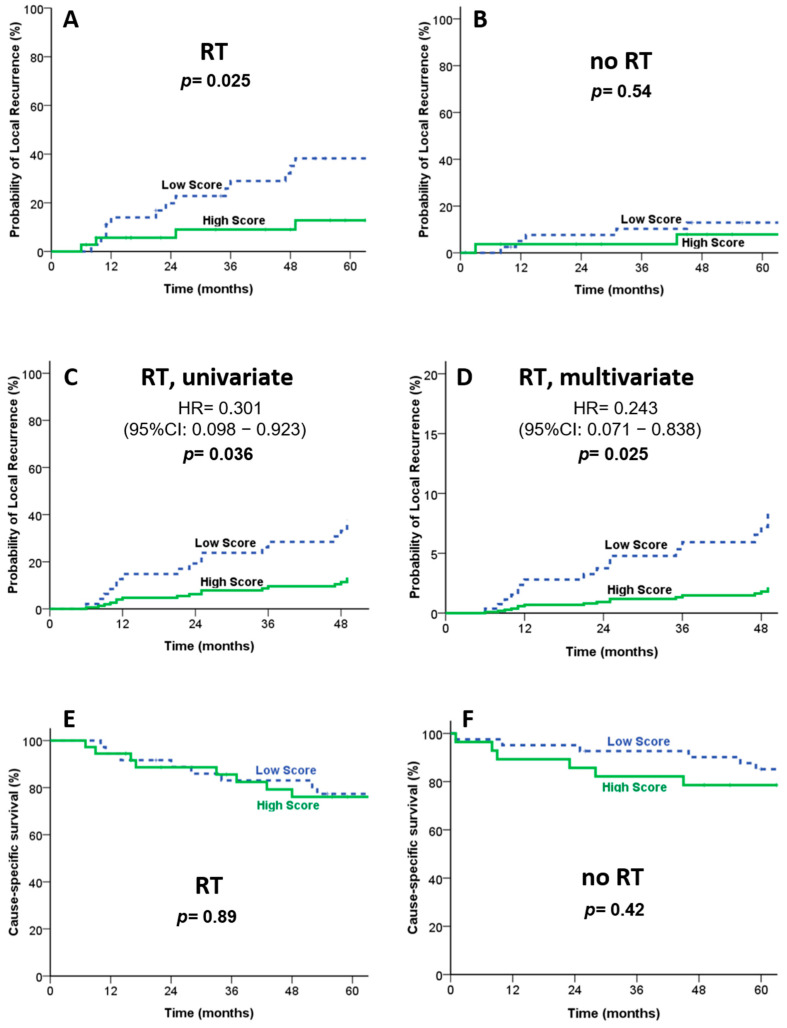
Clinical outcomes according to IRAK2 expression in patients treated with or without RT. (**A**) Kaplan–Meier survival curves on local control for patients treated with RT. (**B**) Kaplan–Meier survival curves on local control for patients treated without RT. (**C**,**D**) Cox proportional hazard regression confirmed that patients with high IRAK2 expression demonstrated better local control than those with a lower expression in univariate (**C**) and multivariate (**D**) analyses. Kaplan–Meier survival curves on cause-specific survival show no statistical difference in the RT (**E**) and no-RT (**F**) groups.

**Table 1 ijms-24-06903-t001:** The top 14 enriched Gene Ontology (GO) terms of the 25 post-irradiation upregulated genes.

GO ID and Term	Count	*p*	Genes
0010647: positive regulation of cell communication	8	0.000190	*CD74*, *ECM1*, *CSF2*, *IRAK2*, *CA2*, *INHBB*, *INHBA*, *HBEGF*
0023056: positive regulation of signaling	8	0.000196	*CD74*, *ECM1*, *CSF2*, *IRAK2*, *CA2*, *INHBB*, *INHBA*, *HBEGF*
0042221: response to chemical	11	0.000696	*C4B*, *CD74*, *HSD3B7*, *ECM1*, *FABP3*, *CSF2*, *IRAK2*, *CA2*, *INHBB*, *INHBA*, *HBEGF*
0006950: response to stress	10	0.001559	*C4B*, *CD74*, *ECM1*, *CSF2*, *IRAK2*, *CA2*, *MCAM*, *INHBB*, *INHBA*, *HBEGF*
0045937: positive regulation of phosphate metabolic process	6	0.001669	*CD74*, *FABP3*, *CSF2*, *IRAK2*, *INHBB*, *INHBA*
0030334: regulation of cell migration	5	0.002614	*CD74*, *ECM1*, *MCAM*, *PHLDA2*, *HBEGF*
2001233: regulation of apoptotic signaling pathway	4	0.004272	*CD74*, *CSF2*, *INHBB*, *INHBA*
0002682: regulation of immune system process	6	0.005518	*C4B*, *CD74*, *ECM1*, *IRAK2*, *CA2*, *INHBA*
0048608: reproductive structure development	4	0.005576	*CSF2*, *INHBB*, *INHBA*, *PHLDA2*
0033993: response to lipid	5	0.005924	*FABP3*, *CSF2*, *IRAK2*, *CA2*, *INHBA*
0001959: regulation of cytokine-mediated signaling pathway	3	0.007031	*CD74*, *ECM1*, *IRAK2*
1901701: cellular response to oxygen-containing compound	5	0.007289	*CSF2*, *IRAK2*, *CA2*, *INHBB*, *INHBA*
0006952: defense response	6	0.008326	*C4B*, *CD74*, *ECM1*, *IRAK2*, *INHBB*, *INHBA*
0042127: regulation of cell proliferation	6	0.009138	*CD74*, *ECM1*, *FABP3*, *CSF2*, *INHBA*, *HBEGF*

**Table 2 ijms-24-06903-t002:** Clinicopathological features of 172 oral cancer patients according to the expression level of IRAK2.

	IRAK2				*p*-Value
	Low-Expressed (*n* = 87)		High-Expressed (*n* = 85)		
**Age**					
Age ≦ 50	32	36.8%	32	37.6%	0.91
Age > 50	55	63.2%	53	62.4%	
**Gender**					
Male	76	87.4%	80	94.1%	0.13
Female	11	12.6%	5	5.9%	
**Pathology grade**					
G1	8	9.2%	4	4.7%	0.51
G2	73	83.9%	75	88.2%	
G3	6	6.9%	6	7.1%	
**pT**					
pT1	39	44.8%	20	23.5%	0.01
pT2	32	36.8%	34	40.0%	
pT3	6	6.9%	10	11.8%	
pT4	10	11.5%	21	24.7%	
**pN**					
pN0	73	83.9%	61	71.8%	0.14
pN1	7	8.0%	14	16.5%	
pN2	7	8.0%	10	11.8%	
**Pathological stage**					
I	36	41.4%	19	22.4%	0.02
II	26	29.9%	24	28.2%	
III	11	12.6%	15	17.6%	
IVA-IVB	14	16.1%	27	31.8%	
**RT**					
No	44	50.6%	35	41.2%	0.22
Yes	43	49.4%	50	58.8%	
**CT**					
No	51	58.6%	43	50.6%	0.29
Yes	36	41.4%	42	49.4%	
**CCRT**					
No	64	73.6%	53	62.4%	0.12
Yes	23	26.4%	32	37.6%	
**Treatment**					
OP alone	40	46.0%	30	35.3%	0.38
OP + RT	20	23.0%	18	21.2%	
OP + RT/CT	4	4.6%	5	5.9%	
OP + CCRT	23	26.4%	32	37.6%	
**Margin status**					
Margin < 1 mm	11	12.6%	20	23.5%	0.06
Margin ≧ 1 mm	76	87.4%	65	76.5%	
**Bone invasion**					
Negative	81	93.1%	67	78.8%	0.01
Positive	6	6.9%	18	21.2%	
**Extracapsular spread of lymph node**					
Negative	85	97.7%	80	94.1%	0.28
Positive	2	2.3%	5	5.9%	
**Lymphatic permeation**					
Negative	82	94.3%	74	87.1%	0.10
Positive	5	5.7%	11	12.9%	
**Vascular permeation**					
Negative	86	98.9%	80	94.1%	0.12
Positive	1	1.1%	5	5.9%	
**Perineural invasion**					
Negative	73	83.9%	70	82.4%	0.79
Positive	14	16.1%	15	17.6%	
**Submandibular gland invasion**					
Negative	87	100.0%	83	97.6%	0.24
Positive	0	.0%	2	2.4%	
**Skin invasion**					
Negative	83	95.4%	82	96.5%	0.72
Positive	4	4.6%	3	3.5%	

Abbreviations: pT, pathological tumor; pN, pathologic lymph node; RT, radiotherapy; CT, chemotherapy; CCRT, concurrent chemoradiotherapy; OP, operation.

**Table 3 ijms-24-06903-t003:** Univariate (crude) and multivariate (adjusted) logistic regression analysis.

	Crude Estimate (Univariate)					Adjusted Estimate (Multivariate)				
	Coefficient	HR	95% CI		*p*-Value	Coefficient	HR	95% CI		*p*-Value
IRAK2 (Low, ref)	−0.90	0.41	0.16	1.02	0.056	−1.33	0.26	0.10	0.71	0.008
Age (Continuous)	−0.01	0.99	0.96	1.02	0.52	−0.02	0.98	0.94	1.02	0.33
Gender (Male, ref)	−0.33	0.72	0.17	3.06	0.66	−0.29	0.75	0.17	3.28	0.70
Grade 1 (ref)										
Grade 2	0.52	1.69	0.23	12.51	0.61	−0.17	0.85	0.10	7.04	0.88
Grade 3	−12.59	0.00	0.00		0.98	−12.56	0.00	0.00		0.98
pStage I (ref)										
pStage II	−0.31	0.73	0.26	2.06	0.55	0.29	1.34	0.45	3.95	0.60
pStage III	0.84	2.31	0.86	6.22	0.10	1.26	3.52	1.00	12.41	0.05
pStage IV	0.51	1.66	0.36	7.71	0.51	0.88	2.40	0.39	14.81	0.35
OP (ref)										
OP with adjuvant RT	1.29	3.64	1.24	10.67	0.018	1.42	4.16	1.32	13.07	0.015
OP with adjuvant CT	1.54	4.66	0.90	24.03	0.07	1.60	4.94	0.78	31.33	0.09
OP with adjuvant CCRT	1.04	2.84	0.90	8.94	0.08	0.09	1.09	0.26	4.52	0.91

Abbreviations: OP, operation; RT, radiotherapy; CT, chemotherapy; CCRT, concurrent chemoradiotherapy.

**Table 4 ijms-24-06903-t004:** Comparison of 5-year overall and cancer-cause-specific survival in correlation with pathology stage and expression level of IRAK2.

		5-Year Overall Survival				5-Year Cancer-Cause-Specific Survival			
		RT Group (%)	*p*-Value	Non-RT Group (%)	*p*-Value	RT Group (%)	*p*-Value	Non-RT Group (%)	*p*-Value
Pathology Stage I–II	Low IRAK2	80.8	0.64	88.9	0.15	88.0	0.90	91.5	0.306
	High IRAK2	73.7		75		89.2		83.3	
Pathology Stage III–IVA/B	Low IRAK2	58.8	0.63	50.0	0.63	58.8	0.72	50.0	0.633
	High IRAK2	51.6		45.5		64.6		45.5	

## Data Availability

All data supporting the conclusions of this article are available upon request.

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
