# Peer review of "IRAK2, an Immune and Radiation-Response Gene, Correlates with Advanced Disease Features but Predicts Higher Post-Irradiation Local Control in Non-Metastatic and Resected Oral Cancer Patients"

_ijms, 2023, doi:10.3390/ijms24086903_

Round 1

Reviewer 1 Report (New Reviewer)

IRAK2, an immune and radiation-response gene, correlates 2 with advanced disease features but predicts higher 3 post-irradiation local control in non-metastatic and resected 4 oral cancer patients is a paper aimed to  define the clinical role of IRAK2 in mediating tumor response to radiotherapy in172 stage I-IVB resected oral cancer patients. IRAK2 involved ten of 14 top enriched post-irradiation biological processes, focusing on stress response and immune modulation. Clinically, high IRAK2 expression was correlated with adverse disease features, including pT3-4 status (P = 0.01), advanced overall stage (P = 0.02), and positive 29 bone invasion (P = 0.01). In patients who underwent radiotherapy, the IRAK2-high group was associated with fewer post-irradiation local recurrence (P = 0.025) than the IRAK2-low group. The Authors concluded that IRAK2 plays a crucial role in the radiation-induced response. Patients with high IRAK2 32 expression demonstrated more advanced disease features but predicted higher post-irradiation local control in clinical. These findings support IRAK2 as a potential predictive biomarker for radiotherapy response in non-metastatic and resected oral cancer patients.

The median IRAK2 value was applied as a cutoff value to differentiate high 228 or low expression. In order to verify the prognostic value of IRAK2, I would suggest to the Authors to identify the cut-off value using a ROC curve with the Youden index and its associated criterion value and  possibly to performt the survival analysis using the identified cut-off value.

I would suggest to the Authors to analyze the survival (cancer cause-specific survival and overall survival in both the RT and no-RT groups) in relation to the pathological stage and to compare for each stage Low vs High IRAK 2 expression.

Author Response

Dear Editors and Reviewers: 

Enclosed, please find the revised manuscript entitled " IRAK2, an immune and radiation-response gene, correlates with advanced disease features but predicts higher post-irradiation local control in non-metastatic and resected oral cancer patients" that we re-submitted to the International Journal of Molecular Sciences as an Original Research Article.

First, thank you for inviting us to submit a revision of the present work. We replied to all comments on the following pages accordingly. All revised sites in the present manuscript were marked with red color.

All authors have read and approved the re-submitted manuscript. The revised manuscript is significant enough to meet the scope of the International Journal of Molecular Sciences. Thank you for your attention, and we look forward to hearing good news from you soon.

Yours Sincerely,

Shih-Kai Hung, M.D. Ph.D.

Reviewer 2 Report (New Reviewer)

The study is interesting and the manuscript is well-written and designed.

It is recommended to express the aim in a more clear way in the abstract and the introduction section.

Some abbreviations were not defined at first mention in the manuscript. A careful revision of this point is needed.

It is recommended that the authors define in a more clear way the limitations of the study and demonstrate their experience from this study for helping other researchers to develop this study to a higher level.

Author Response

Dear Editors and Reviewers: 

Enclosed, please find the revised manuscript entitled " IRAK2, an immune and radiation-response gene, correlates with advanced disease features but predicts higher post-irradiation local control in non-metastatic and resected oral cancer patients" that we re-submitted to the International Journal of Molecular Sciences as an Original Research Article.

First, thank you for inviting us to submit a revision of the present work. We replied to all comments on the following pages accordingly. All revised sites in the present manuscript were marked with red color.

All authors have read and approved the re-submitted manuscript. The revised manuscript is significant enough to meet the scope of the International Journal of Molecular Sciences. Thank you for your attention, and we look forward to hearing good news from you soon.

Yours Sincerely,

Shih-Kai Hung, M.D. Ph.D.

This manuscript is a resubmission of an earlier submission. The following is a list of the peer review reports and author responses from that submission.

Round 1

Reviewer 1 Report

The manuscript “IRAK2, an Immune and Radiation-Response Gene, Correlates with Advanced Disease Features but Predicts High Post-Irradiation Local Control in Resected Oral Cancer Patients” conducted gene ontology to explore the functional biological analysis of IRAK2 and performed case analysis to define its clinical role in disease progression and mediating tumor response to RT. The group conducted a prior study (Yu et al 2021) where they selected candidate genes responsible for radio-sentitivity identifying the IRAK2 gene. Their previous work found that IRAK2 expresses low levels in radioresistant cell lines, whereas overexpression of it restored the radiosensitivity (via caspase 8/3-mediated apoptosis), suggesting this gene´s involvement in tumor response to ionizing radiation. In their previous study, the authors sequenced, selected the candidate gene, overexpressed, knocked it down, performed colony formation, Western Blotting, qRT-PCR, Flow Cytometry, Immunohistochemistry as well as in vivo tumorigenesis. The previous study even evaluated the expression of IRAK2 in 41 OSCC patient samples. Therefore, I was not able to find what is novel to this manuscript. When I read this manuscript I thought it was a really good study, well conducted and written, until I read their other study (Yu et al 2021) and could not separate the objectives among them. My suggestion is to focus on what´s novel on this research, leave what was already published and present only what is new.

Reviewer 2 Report

The present work addresses a topic of importance for clinical and basic research –biological analysis of IRAK2 and its role in the radioresistance in both OSCC patients and cell.

Abstract is well structured and informative. Introduction is very short and needs extension. The authors could include information concerning the world burden of OSCC, duration of life, therapy and etc.  The results are satisfactory.  Their presentation is logically developed. Data shown are adequate to the aim.

Some important remarks are listed below:

1. Lines 26-27 “For identified significant IR-responsive genes, gene ontology (GO) analysis was conducted for exploring their involved bio-logical processes.” The sentence needs rewriting.

2. Lines 88-89 “Gene Ontology (GO) analysis focused on these genes, for which the marked changes may predict biological processes.” The sentence needs rewriting.

3. Why only IRAK2 was used for analysis? In addition to IRAK2, CD74 is much stronger induced and it is involved in 11 out of 14 GO terms.

4. Does the IRAK2 expression is able to distinguish radio-sensitive from radio-resistant OSCC patients? In this aspect there would be interesting to design  ROC curves in order to evaluate IRAK2 biomarker properties.

5. Lines 154 NSCLC needs abbreviation.   

6. The conclusion part is very short and needs great expanding.

In conclusion, the topic of the manuscript is very interesting and will stimulate the reader’s interest. Finally, I recommend the proposed manuscript to be accepted for publication at IJMS  after answering the above mentioned corrections.

Round 2

Reviewer 1 Report

The manuscript “IRAK2, an Immune and Radiation-Response Gene, Correlates with Advanced Disease Features but Predicts High Post-Irradiation Local Control in Resected Oral Cancer Patients” conducted gene ontology to explore the functional biological analysis of IRAK2 and performed case analysis to define its clinical role in disease progression and mediating tumor response to RT. However, the study does not present a methodology consistent with the results presented. The authors included cell culture analysis, western blotting and immunohistochemistry in their methodology, but did not present any results from them. Furthermore, due to the nature of the results, it may be more appropriate to direct the findings to a journal focused on bioinformatics. All other results were already presented in their previous manuscript.